# The *Paramecium* histone chaperone Spt16-1 is required for Pgm endonuclease function in programmed genome rearrangements

Augustin de Vanssay[1], Amandine Touzeau[1], Olivier Arnaiz[2], Andrea Frapporti[1], Jamie Phipps[1], Sandra Duharcourt[1]*

**1** Université de Paris, Institut Jacques Monod, CNRS, Paris, France, **2** Université Paris-Saclay, CEA, CNRS, Institute for Integrative Biology of the Cell (I2BC), Gif-sur-Yvette, France

☯ These authors contributed equally to this work.
¤ Current address: Wellcome Trust/Cancer Research UK Gurdon Institute, University of Cambridge, Cambridge, United Kingdom
* sandra.duharcourt@ijm.fr

**Data Availability Statement:** All relevant data are within the manuscript and its Supporting Information files.

## Abstract

In *Paramecium tetraurelia*, a large proportion of the germline genome is reproducibly removed from the somatic genome after sexual events via a process involving small (s) RNA-directed heterochromatin formation and DNA excision and repair. How germline limited DNA sequences are specifically recognized in the context of chromatin remains elusive. Here, we use a reverse genetics approach to identify factors involved in programmed genome rearrangements. We have identified a *P. tetraurelia* homolog of the highly conserved histone chaperone Spt16 subunit of the FACT complex, Spt16-1, and show its expression is developmentally regulated. A functional GFP-Spt16-1 fusion protein localized exclusively in the nuclei where genome rearrangements take place. Gene silencing of Spt16-1 showed it is required for the elimination of all germline-limited sequences, for the survival of sexual progeny, and for the accumulation of internal eliminated sequence (ies) RNAs, an sRNA population produced when elimination occurs. Normal accumulation of 25 nt scanRNAs and deposition of silent histone marks H3K9me3 and H3K27me3 indicated that Spt16-1 does not regulate the scanRNA-directed heterochromatin pathway involved in the early steps of DNA elimination. We further show that Spt16-1 is required for the correct nuclear localization of the PiggyMac (Pgm) endonuclease, which generates the DNA double-strand breaks required for DNA elimination. Thus, Spt16-1 is essential for Pgm function during programmed genome rearrangements. We propose a model in which Spt16-1 mediates interactions between the excision machinery and chromatin, facilitating endonuclease access to DNA cleavage sites during genome rearrangements.

## Author summary

The genome is generally similar in all the cells of an organism. However, in the ciliate *Paramecium tetraurelia*, massive and reproducible programmed DNA elimination leads to a

**Funding:** This work was supported by intramural funding from the CNRS, the Fondation de la Recherche Médicale (Equipe FRM DEQ20160334868), Labex Who Am I? #ANR-11-LABX-0071 and the Université de Paris IdEx #ANR-18-IDEX-0001 funded by the French Government through its "Investments for the Future" program, the Agence Nationale de la Recherche (ANR-18-CE12-0005-03 ; ANR-19-CE12-0015-01). A. de V. was recipient of a postdoctoral fellowship from 'Fondation ARC' and A.T. was recipient of PhD fellowships from 'Ministère de l'Enseignement Supérieur et de la Recherche' and 'Fondation ARC'. A. F. was recipient of a LABEX Who Am I? transition postdoc fellowship. We acknowledge the ImagoSeine facility, member of the France BioImaging infrastructure supported by the ANR-10-INSB-04. The funders had no role in study design, data collection and analysis, decision to publish, or preparation of the manuscript.

**Competing interests:** The authors have declared that no competing interests exist.

highly streamlined somatic genome. In eukaryotes, DNA is packaged into nucleosomes, which ensure genome integrity but act as a barrier to enzymes acting on DNA. How the endonuclease PiggyMac gains access to the genome to initiate DNA elimination remains elusive. Here, we identified four *P. tetraurelia* genes encoding homologs of the conserved histone chaperone Spt16, which can modulate access to DNA by promoting nucleosome assembly and disassembly. We demonstrated that the most divergent gene, *SPT16-1*, has a highly specialized expression pattern, similar to that of PiggyMac, and a specific role in programmed DNA elimination. We show that the Spt16-1 protein, like PiggyMac, is exclusively localized in the differentiating somatic nucleus, and is also required for the dramatic elimination of germline-limited sequences. We further show that Spt16-1 directs the correct nuclear localization of the PiggyMac endonuclease. Thus, Spt16-1 is essential for PiggyMac function during programmed DNA elimination. We propose that Spt16-1 mediates the interaction between PiggyMac and chromatin or DNA, facilitating endonuclease access to DNA cleavage sites.

## Introduction

In the unicellular eukaryote *Paramecium tetraurelia*, a large proportion of the germline genome is reproducibly removed from the somatic genome after sexual events via a process involving small (s)RNA-directed heterochromatin formation and subsequent DNA excision and repair. The *P. tetraurelia* endonuclease PiggyMac (Pgm) must gain access to the genome to initiate excision of DNA for it to be eliminated. In eukaryotes, DNA is packaged into nucleosomes, which ensure genome integrity but also represent a barrier to enzymes acting on DNA. Histone chaperones can modulate this barrier by promoting nucleosome assembly and disassembly (reviewed in [1]). The conserved histone chaperone FACT, composed of two subunits, Spt16 and Ssrp1/Pob3, is key to maintaining the structural and regulatory functions of chromatin [2,3]. *In vitro*, FACT engages in multiple interactions with nucleosomal DNA and histones to reorganize nucleosomes and allow access to DNA [4–9].

In *P. tetraurelia* the highly polyploid somatic macronucleus (MAC) mediates gene expression and is destroyed at each sexual cycle [10]. Germline functions are supported by two small, diploid micronuclei (MICs) that are transcriptionally silent during vegetative growth. During sexual events, the MICs undergo meiosis and transmit the germline genome to the zygotic nucleus. New MICs and new MACs develop from mitotic copies of the zygotic nucleus. Development of the new MAC involves massive and highly reproducible genome rearrangements [11]. In addition to the imprecise elimination of large genomic regions, up to several kb in length, containing repeated sequences, 45,000 short, single-copy internal eliminated sequences (IESs) (3.5 Mbp) are precisely excised from intergenic and coding regions [12]. IES excision is initiated by DNA double strand breaks (DSBs) at IES ends, mediated by the domesticated transposase PiggyMac (Pgm) and assisted by Pgm-Like proteins [13,14]. DSBs are repaired precisely by the classical non-homologous repair (NHEJ) pathway [15,16].

The question of how such diverse sequences are recognized and excised remains elusive. Different classes of small RNAs are required for elimination of most germline-limited sequences. The 25 nt-long scanRNAs are produced from the germline genome during meiosis by Dicer-like proteins [17,18], loaded onto Piwi proteins and transported to the maternal MAC, where selection of MIC-specific scanRNAs is thought to take place [19–21]. scanRNAs would then guide the deposition of histone H3 post-translational modifications (H3K9me3 and H3K27me3) onto sequences to be eliminated in the developing MAC and specifically

tether the Pgm elimination machinery to germline sequences [22–25]. A second class of developmental-specific sRNAs, the iesRNAs, 26 and 31 nt in length and which map exclusively to IESs, are likely produced in the developing MAC after excision by another Dicer-like protein and faciliate efficient IES excision [18,26].

In order to identify factors involved in programmed genome rearrangements, we selected *P. tetraurelia SPT16-1*, a homolog of the Spt16 subunit of the FACT complex, because its gene expression, as measured by RNAseq, is significantly up-regulated during MAC development and displays an expression profile similar to that of the *PGM* endonuclease gene. We demonstrated that a functional GFP fusion Spt16-1 protein is specifically localized in the new developing MAC. Resequencing the genome upon *SPT16-1* knockdown (KD) revealed that Spt16-1 is required for the elimination of all germline specific sequences. *SPT16-1*-KD small RNA sequencing indicated that Spt16-1 is essential for iesRNA accumulation but not for the biogenesis or selection of scanRNAs. *SPT16-1* KD did not affect the deposition of H3K9me3 and H3K27me3 histone marks in the new developing MAC, suggesting that Spt16-1 acts downstream of scanRNA-directed heterochromatin formation. We further showed that Spt16-1 is required for the correct nuclear localization of Pgm in the new developing MAC but not for its expression or stability. Collectively, our data reveal that the *P. tetraurelia* developmental Spt16-1 is a regulator of programmed genome rearrangements. We present a model whereby Spt16-1 is required for the association of the Pgm elimination machinery with chromatin.

## Results

### Identification of *Paramecium tetraurelia* Spt16 and Pob3 homologs

Given human SPT16 has been shown to play a role in chromatin reorganization at the sites of DNA damage [27], we wanted to understand whether SPT16 in organisms that show programmed genome rearrangements might use SPT16 to facilitate this process. To identify putative *Paramecium* Spt16 homologs, BLAST searches against the *P. tetraurelia* somatic genome were performed using the human/yeast SPT16 protein sequences as a query [28]. We identified four genes encoding Spt16-like proteins, grouped in two families (Fig 1). Three proteins (Spt16-2a, -2b and -2c) belong to the same family and are close paralogs, arising from *Paramecium* whole genome duplications [29] (Fig 1A). Spt16-1 is the most evolutionary distant of the Spt16 sequences analyzed, having only 33% amino acid identity with the Spt16-2a, -2b and -2c protein sequences (Fig 1A). In contrast to most other eukaryotes, which have a single Spt16 protein, *Paramecium* species harbor two Spt16 families. In *P. caudatum*, two genes are found, one in each family, while in other *Paramecium* species, multiple paralogs are found in the Spt16-2 family (S1 Fig). In all *Paramecium* species however, Spt16-1 is encoded by a single gene.

All four *P. tetraurelia* Spt16 proteins share the characteristic Spt16 domain organization (Fig 1B): The N domain, composed of a lobe domain and a non-catalytic peptidase M24 domain, which is non-essential but interacts with histones H2A and H3-H4; the dimerization D domain; the M domain homologous to Rtt106, which interacts with H3-H4; and the C domain, which contains the minimal binding domain (MBD), required for H2-H2B interaction [4,6,7,30]. The aromatic and acidic amino acids required for interaction with histones in the MBD are conserved in *P. tetraurelia* (S2 Fig).

The expression of *SPT16* genes during the *P. tetraurelia* life cycle, examined using RNA-Seq [31] revealed that *SPT16*-2a, -2b, and -2c were expressed constitutively during vegetative growth and during the sexual process of auto-fertilization (autogamy) (Fig 1C). In contrast, *SPT16-1* was specifically up-regulated during autogamy, similar to the *PiggyMac* (*PGM)* gene [13], and was not detected during vegetative growth (Fig 1C). *SPT16-1* induction coincided

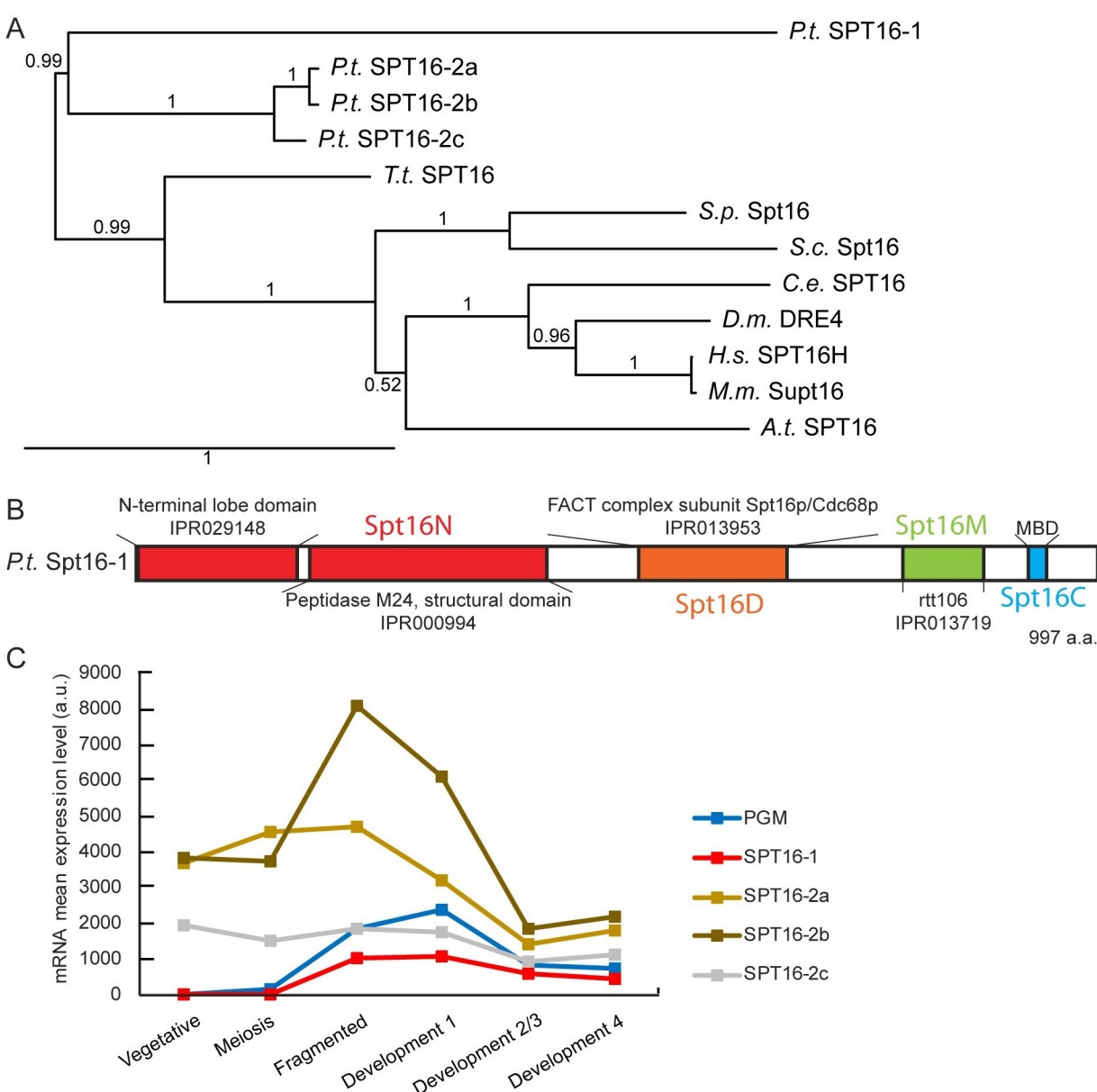

**Fig 1.** *Paramecium tetraurelia* **Spt16 proteins.** **(A)** Phylogenetic tree of Spt16 proteins from *Paramecium tetraurelia* (Pt), *Tetrahymena thermophila* (Tt), *Saccharomyces cerevisiae* (Sc), *Schizosaccharomyces pombe* (Sp), *Caenorhabditis elegans* (Ce), *Drosophila melanogaster* (Dm), mouse (Mm), human (Hs) and *Arabidopsis thaliana* (At) based on the alignment of full-length protein sequences. The tree was generated with PhyML 3.0 with bootstrapping procedure and visualized with Tree.Dyn 198.3. Accession numbers are provided in S1 Table. **(B)** Conserved domains (colored boxes) in *P. tetraurelia* Spt16-1 protein. **(C)** *SPT16* gene expression profiles during the life cycle. Mean mRNA expression levels were determined by RNA sequencing during vegetative growth and at different time points during autogamy [31]. See S1 File.

with development of the new MAC after sexual events. Based on this expression profile, we decided to focus on Spt16-1.

SPT16 is known to form a heterodimer with the SSRP1 (Structure Specific Recognition Protein 1) or Pob3 (Polymerase One Binding protein 3) in yeast [3]. In contrast to the conservation of Spt16 domain architecture in eukaryotes, SSRP1, although also highly conserved in its core domains, differs significantly among unicellular eukaryotes, plants, and metazoans due to the variable inclusion of domains. The yeast SSRP1, Pob3, lacks both C-terminal domains, a

high mobility group (HMG) box and a C-terminal intrinsically disordered domain. To identify putative *Paramecium* Ssrp1/Pob3 homologs, we performed BLAST searches against the *P. tetraurelia* somatic genome. We identified three genes encoding Pob3 proteins, which, like their yeast counterparts, do not contain a HMG box (S3 Fig). As observed for Spt16 proteins, the Pob3 proteins group in two families (S3 Fig): (i) one family with two close paralogs (Pob3-2a and -2b) arising from *Paramecium* whole genome duplications [29], which are expressed constitutively during vegetative growth and during autogamy (S3 Fig); and (ii) another family with one Pob3-1 protein, which shares only 64% identity with the two other Pob3 proteins. The expression of the *POB3-1* gene was found to be specifically upregulated during development and displayed a very similar expression pattern to *SPT16-1*, suggesting that they could be acting together within a developmental specific FACT complex (S3 Fig). However, we noticed that, even though the three *P. tetraurelia* Pob3 proteins share the characteristic Pob3 domain organization (the SSRP1 and histone chaperone Rttp106-like domains, which contain two PH motifs, similar to SPT16 M domain, and are likely to be involved in binding histones), they lack the N-terminal N domain sequence (S4 Fig), which is responsible for dimerization with SPT16 protein in yeast [32].

## Spt16-1 is required for post-zygotic development

To test whether Spt16-1 is required during sexual events, we knocked-down *SPT16-1* expression (*SPT16-1* KD) during autogamy, using RNA interference (RNAi) induced by feeding *Paramecium* with *E. coli* producing Spt16-1 dsRNA [33] (Fig 2 and S5 Fig). Autogamy commenced normally in *SPT16-1* KD cells, which eventually formed two new developing MACs. However, post-autogamous cells were unable to resume vegetative growth when returned to normal medium, as observed upon *PGM* silencing (Fig 2A). Post-autogamous cells displayed two large macronuclei and died before the first cellular division, similar to *PGM* silenced cells (Fig 2B). To control for off-target effects, two non-overlapping fragments of the *SPT16-1* gene were used to generate constructs for *SPT16-1* RNAi. Both constructs gave similar results (6% of sexual progeny are viable) (Fig 2A). We perfomed similar knock-down experiments during autogamy using two non-overlapping fragments of the *POB3-1* gene for RNAi. In contrast to *SPT16-1* KD, we were not able to observe any lethality in the sexual progeny after *POB3-1* KD (S3 Fig), raising the possibility that *SPT16-1* and *POB3-1* exert distinct functions.

The lethality observed in sexual progeny in the *SPT16-1* KD could be due to defect(s) in events that occur during autogamy: MIC meiosis, karyogamy, and/or new MAC development. Given *SPT16-1* expression induction corresponds to the time when new MACs are developing, we initiated *SPT16-1* KD at the onset of MAC development using conjugation between mating partners (Fig 2C and S1 File) [34]. Strong lethality was observed in the *SPT16-1* KD, as for the *PGM* KD, while no effect was observed in the RNAi control (Fig 2C). We conclude that Spt16-1, like Pgm, is essential for MAC development and production of viable sexual progeny.

## Spt16-1 protein localizes in developing new somatic macronuclei

To understand Spt16-1 function, we examined the subcellular localization of the Spt16-1 protein. We constructed an RNAi-resistant *GFP-SPT16-1* transgene, in which GFP was inserted immediately downstream of the *SPT16-1* start codon (Materials and Methods). Transgene expression was under the control of *SPT16-1* up- and downstream sequences. No detectable GFP fluorescence was observed in vegetative cells or during the sexual events of autogamy, following transgene microinjection into the MAC of vegetative cells (Fig 3A). GFP fluorescence accumulated in the new developing MAC during autogamy, consistent with *SPT16-1* mRNA

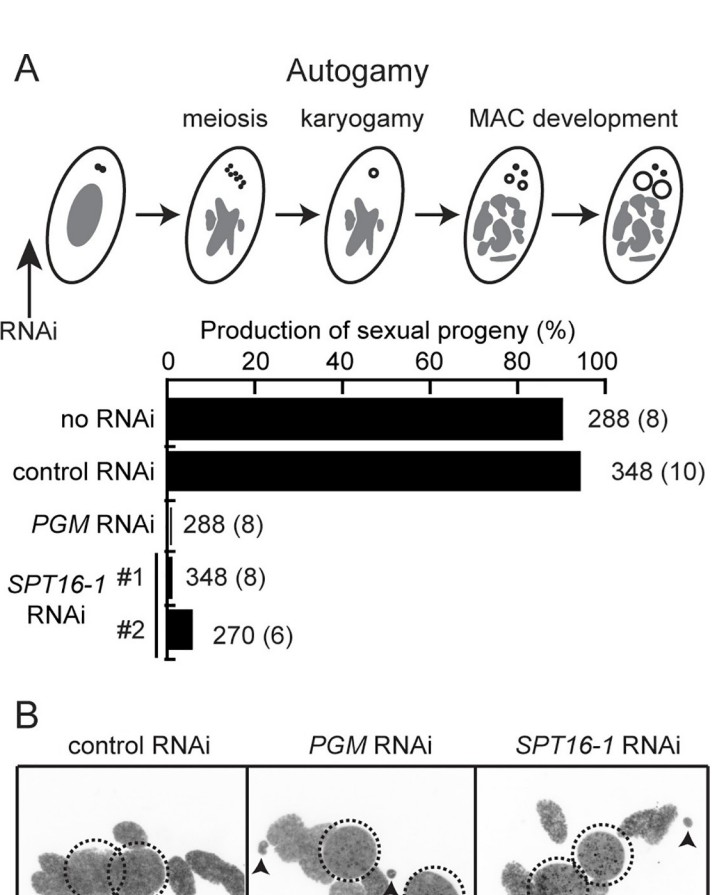

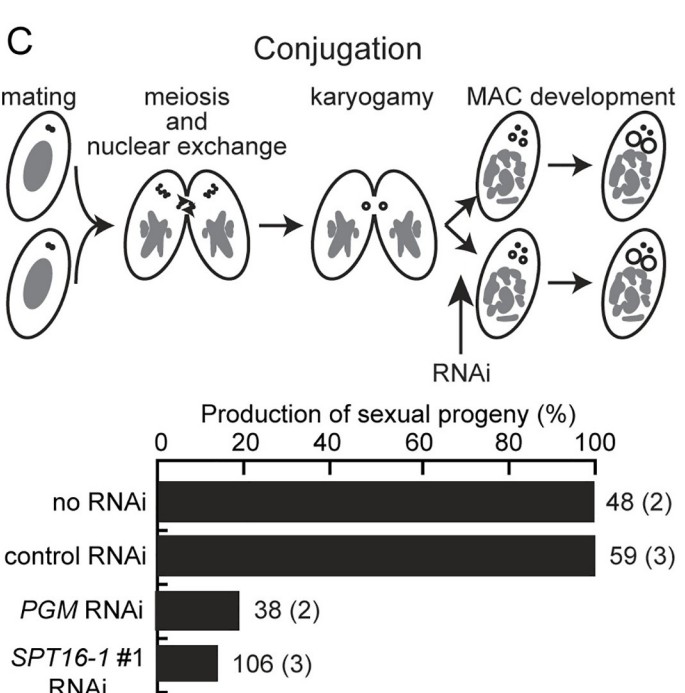

**Fig 2. Spt16-1 is essential for development. (A)** Production of post-autogamous sexual progeny following *SPT16-1* gene silencing. The gene targeted in each silencing experiment is indicated. Two non-overlapping silencing fragments (#1 and #2) of the *SPT16-1* gene were used independently. The *ND7* or *ICL7* genes were used as control RNAi targets, since their silencing has no effect on sexual processes [21]. The total number of autogamous cells analyzed for each RNAi and the number of independent experiments (in parenthesis) are indicated. Death in progeny after *SPT16-1* silencing was observed after less than three cell divisions, as for *PGM* silencing. **(B)** Z-projections of Hoechst staining of control, Pgm or Spt16-1-depleted cells fixed at late stages of development during autogamy. Dashed black circles indicate the two developing MACs. Black arrows indicate the MICs. The other Hoechst-stained nuclei are fragments from the maternal somatic MAC. Scale bar is 10 μm. **(C)** *SPT16-1* gene silencing following mating pair separation. See S1 File. The gene targeted in each silencing experiment is indicated. Only silencing fragment #1 of the *SPT16-1* gene was used. The total number of cells analyzed for each RNAi and the number of independent experiments (in parenthesis) are indicated.

expression (Fig 3A). The GFP-Spt16-1 fusion protein was functional, as it was able to rescue lethality in the post-autogamous progeny of injected cells, following an *in vivo* complementation assay (Fig 3B). In non-transformed cells, *SPT16-1* KD induced substantial lethality (16% of viable progeny) in sexual progeny, indicating effective silencing of the endogenous *SPT16-1* gene. *SPT16-1* KD induction in the *GFP-SPT16-1* transformed cells resulted in recovery of viability of the sexual progeny (83%), indicating the fusion protein complemented the lethality caused by depletion of the endogenous Spt16-1 protein (Fig 3B). The GFP-Spt16-1 fusion protein is detected in new developing MACs of cells inactivated for *SPT16-1* (S6 Fig), indicating the fusion encodes a functional protein exclusively localized in the developing MAC.

## Spt16-1 is essential for programmed genome rearrangements

To analyze the role of Spt16-1 during MAC development, we asked whether it is required for programmed genome rearrangements. To investigate *SPT16-1* function in IES excision, genomic DNA was extracted after *SPT16-1*, *PGM*, or control silencing, at a time when IES excision is normally finished, and analyzed by PCR. In control RNAi experiments, the 7 IESs assayed were completely excised from the new developing MACs. In contrast, IES-retaining forms accumulated in the new MACs of *PGM* or *SPT16-1* KD cells (Fig 4A). Similar results were obtained for both *SPT16-1* silencing constructs (Fig 4A). The amplification of non-rearranged DNA indicated programmed genome rearrangements did not proceed normally after Spt16-1 depletion. Because of the presence of rearranged forms in the fragments of the maternal MAC, it is difficult to assess whether new excision junctions are formed in the new developing MAC using this approach.

To circumvent this issue, we used a cell line (delta A) that has a wild-type MIC genome, but carries a maternally inherited deletion of the non-essential A gene in the MAC. During autogamy, all IESs are normally excised in this cell line before the entire locus is deleted in the new MAC. Using PCR primers in flanking sequences of two IESs located in the A locus, transient excision junctions corresponding to *de novo* IES excision could be detected (Fig 4B). In contrast, no excision junctions were detected in *SPT16-1* KD cells (Fig 4B). To confirm that excision is impaired in Spt16-1 depleted cells, we monitored the appearance of excised IES circular molecules during autogamy. Covalently closed circles are formed from long excised linear IESs, following Pgm-dependent DNA cleavages at each IES boundary [35]. Using divergent PCR primers internal to IES 51A4578, we transiently detected IES circles in control RNAi, but failed to detect any in cells depleted for Spt16-1 (Fig 4C). Thus, Spt16-1 is required for the recovery of both chromosomal junctions and excised IES circles. We noted that the non-rearranged forms of the A locus accumulated at late time points during autogamy in *SPT16-1* KD cells (Fig 4B), suggesting maternally inherited deletion of the A gene is blocked, and the non-

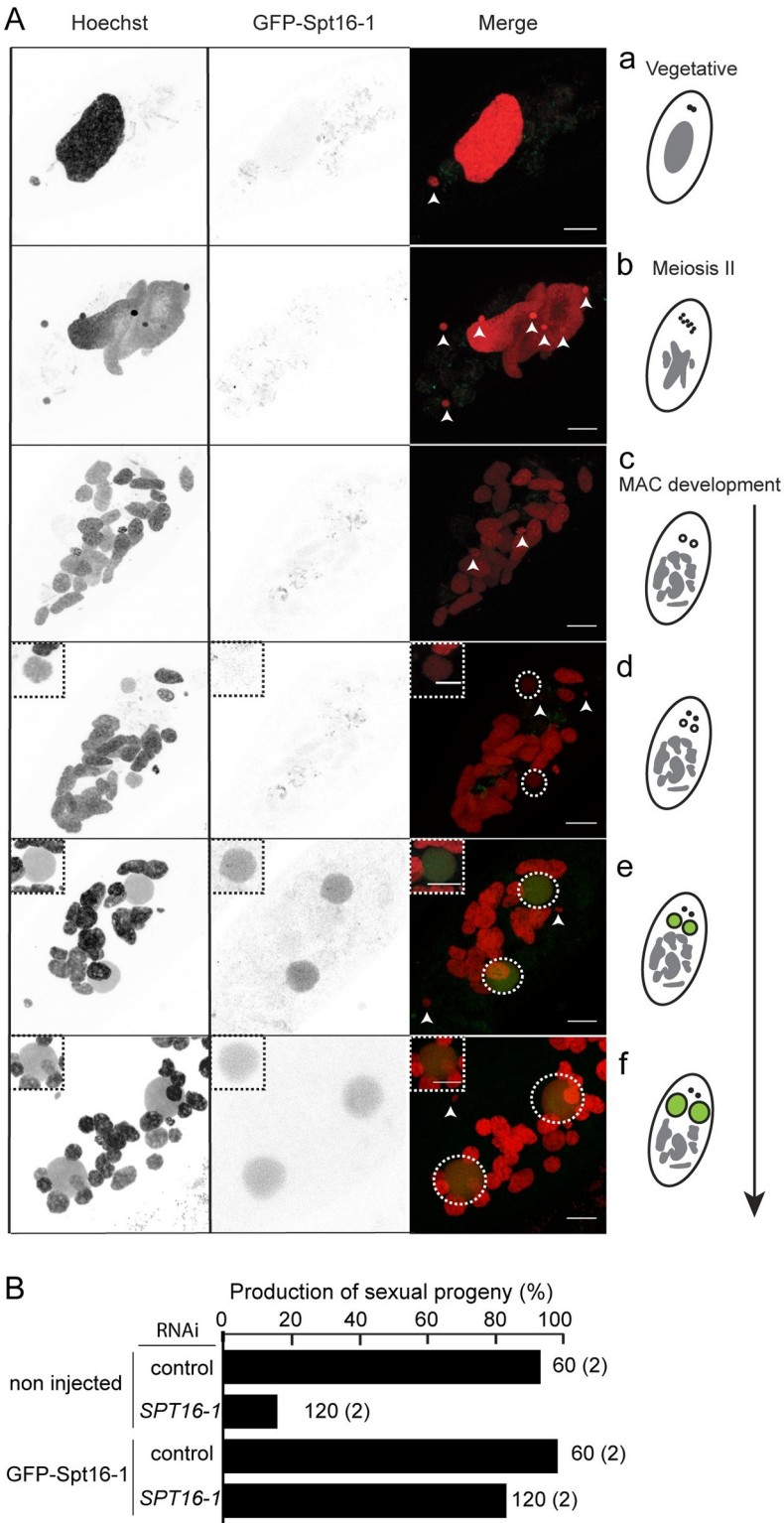

**Fig 3. A functional GFP-Spt16-1 fusion protein is localized in the developing macronuclei.** (A) Localization of the GFP-Spt16-1 fusion protein expressed from the RNAi-resistant *GFP-SPT16-1* transgene (see Materials and Methods). Overlay of Z-projections of magnified views of GFP-Spt16-1 (in green) and Hoechst (in red) are presented (a-f), aligned with their schematic representations on the right. Dashed white circles indicate the two developing MACs. The other Hoechst-stained nuclei are fragments from the maternal somatic MAC or the MICs. Scale bar is 10 μm. The inset

displays one the two developing new MACs. **(B)** *In vivo* genetic complementation with the RNAi-resistant *GFP-SPT16-1* transgene upon *SPT16-1* gene silencing (using SPT16-1#2 RNAi construct) during autogamy. Production of post-autogamous sexual progeny is indicated. The gene targeted in each silencing experiment is indicated. Silencing is performed in non-injected cells or GFP-Spt16-1 injected cells. *ICL7* gene is used as a control RNAi target [21]. The total number of autogamous cells analyzed for each RNAi and the number of independent experiments (in parenthesis) are indicated. See S1 File.

rearranged germline locus is retained in the developing new MACs, a phenotype similar to that observed when DNA cleavage is inhibited [13,16].

To analyze the genome-wide effects of Spt16-1 depletion, we performed high-throughput sequencing of DNA extracted from a nuclear preparation enriched for new MACs of *SPT16-1* KD autogamous cells (S3 Table). To quantify the effects of *SPT16-1* KD on IES retention, we used sequencing data from non-silenced autogamous cells of the same strain, as a control [24]. Excision of all IESs is altered and 98.6% (44,334) IESs are statistically significantly retained in the developing MAC after *SPT16-1* KD, similar to the 99% (44,491) observed after *PGM* KD [36] (S7 Fig). We measured IES retention at each IES boundary, using the boundary score method [36]. The score was 0 in the control, as all IESs are correctly excised. Similar Gaussian distributions for each boundary are observed, with a mean retention of 0.44, upon *SPT16-1* KD, indicating that retention of two boundaries is highly correlated, as seen for *PGM* KD (Fig 4D). Excision is an efficient, robust process but rare excision errors can be detected [37]. We catalogued error events (deletion of a somatic DNA fragment, use of an alternative boundary during IES elimination) [36] and found no significant differences between *SPT16-1* KD, *PGM* KD, or control cells (S7 Fig).

To determine the effects of Spt16-1 depletion on the retention of germline-limited sequences other than IESs, we compared the sequencing read coverage of different genomic compartments in *SPT16-1 KD* and *PGM* KD [11] (Fig 4E). The germline genome that is collinear with MAC chromosomes ("MAC-destined") is similarly covered in all datasets (Fig 4E). In contrast, the germline-limited (MIC-limited) portion of the genome, which contain repeated sequences including transposable elements, is not covered by MAC reads and is well-covered by MIC reads, *PGM* KD, and *SPT16-1* KD reads, indicating that MIC-limited sequences are retained in *SPT16-1* KD cells, as in *PGM* KD cells (Fig 4E). We assessed retention by mapping reads from each dataset to the cloned copies of the Sardine transposon family and found that all copies are retained after *SPT16-1* KD (Fig 4F). Thus, Spt16-1, like Pgm, is essential for all DNA elimination events.

## Spt16-1 is required for iesRNA but not for scanRNA accumulation

To determine at which step Spt16-1 acts in DNA elimination, we examined whether *SPT16-1* KD affected sRNA levels. We used high throughput sRNA sequencing to compare sRNA populations present in control and *SPT16-1* KD cells at three time-points during autogamy—very early (~20% cells undergoing meiosis), early (10h after the first time point), and late (20 h after the first time point) (Fig 5). All sRNA reads obtained for *SPT16-1* KD and a control KD were mapped to reference genomes and read counts were normalized to the total number of reads (See Materials and Methods). scanRNAs are 25-nt RNAs produced from the MIC at very early stages of autogamy and are continuously present until later stages [17,18]. In the very early time point, 25 nt-scanRNAs form the majority of sRNAs in the control sample, as reported previously [17,18,38]. The same is true in *SPT16-1* KD cells: 25-nt scanRNAs are produced normally as shown by a similar number of reads matching the genome in both *SPT16-1* and control RNAi, indicating that Spt16-1 is not required for scanRNA accumulation. As MAC development proceeds in control and *SPT16-1* KD cells, the proportion of scanRNAs

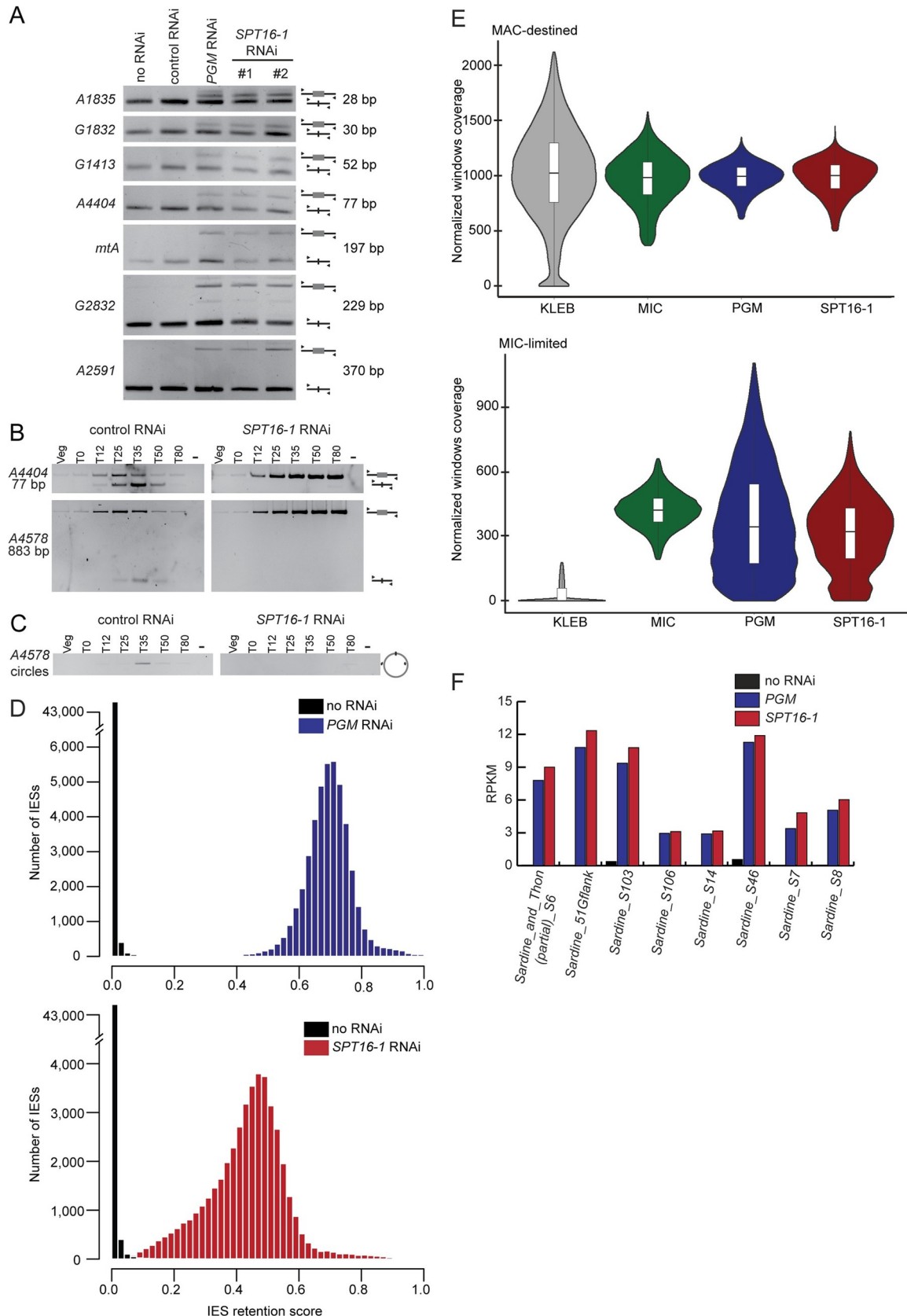

**Fig 4. Spt16-1 is required for DNA elimination events. (A)** PCR analysis of IES retention. Primers (black arrows, S2 Table) are located on either side of the IES. Total DNA samples were prepared from autogamous cell population upon RNAi-mediated silencing of the indicated genes. Because the maternal MAC is still present at this stage, the excised version is amplified in all cases; the IES-retaining fragment can be detected only if it accumulates in the zygotic developing MACs. **(B)** Detection of IES excision junctions in the variant cell line delta A using primers (S2 Table) located on either side of two IESs in the A gene. In vegetative cells, only the MIC contributes to the PCR signal. When new MACs develop, *de novo* chromosomal junctions are detected upon IES excision. At later stages, this signal disappears due to maternal epigenetic inheritance of the A deletion. **(C)** Detection of IES 51A4578 circle junctions by PCR in the variant cell line delta A, using two internal divergent primers (same time-course as in (B)). **(D)** Distribution of IES retention scores after *PGM* (sample from [12], blue, top panel) and *SPT16-1* (red, bottom panel) RNAi, and control (no RNAi) (sample from [24], black). Absolute values of retentions scores cannot be compared between *PGM* KD and *SPT16-1* KD, due to variable contamination by old MAC fragments. **(E)** Violin plot superimposed with a boxplot of normalized coverage using DESeq2 (see [11]) of 1-kb windows of the MIC assembly for a *Klebsiella* (KLEB grey) sample, MIC sample (green), *PGM* RNAi sample (blue) and *SPT16-1* RNAi sample. The windows are split into two categories defined in [11]: MAC-destined compartment and MIC-limited compartment. **(F)** Retention of some class Il DNA transposons imprecisely eliminated after *PGM* and *SPT16-1* RNAi. The bar plots represent read coverage of 8 individual copies of the Sardine transposon (GenBank Accession No. HE774468-HE774475). The coverage was determined by mapping reads for the control dataset (black), *PGM* silencing (blue) and *SPT16-1* silencing (red) datasets. The normalized units (RPKM) are reads per kilobase of the transposon sequence per million library read mapped against the MAC reference genome. As expected, all transposable elements are retained in *PGM* RNAi [11,12].

corresponding to MAC sequences decreases, and consequently the proportion of scanRNAs corresponding to germline-specific sequences increases under both conditions, indicating that selection of MIC-specific scanRNAs occurs normally in *SPT16-1* KD cells (Fig 5).

In the course of development, while the total 25 nt scanRNA population is decreasing, 26–30 nt iesRNAs matching IESs increase in control RNAi, as expected [18]. In contrast to control KD, *SPT16-1* KD substantially reduces iesRNA production (Fig 5), consistent with iesRNA formation requiring IES excision [26]. Thus, Spt16-1 is required for iesRNA accumulation but does not affect the biogenesis or selection of scanRNAs.

## Spt16-1 is not required for H3K9me3 and H3K27me3 deposition in the developing new MAC

We previously reported that H3K9me3 and H3K27me3 deposition in the developing MAC is mediated by the Enhancer of zeste-like 1 (Ezl1) histone methyltransferase and required for DNA elimination [24,25]. To determine whether Spt16-1 is implicated in histone H3 post-translational modifications deposition in the developing MAC, we performed indirect immunostaining experiments using specific H3K9me3- and H3K27me3- antibodies in control and *SPT16-1* KD cells during autogamy. As previously reported, H3K9me3 and H3K27me3 signals were detected in new developing MACs with a diffuse pattern that gradually formed nuclear foci before the signals eventually coalesce in one single nuclear focus and disappear in control RNAi (Fig 6 and S8 Fig) [24]. Depletion of Spt16-1 did not affect the deposition of H3K9me3 and H3K27me3 in the developing MACs (Fig 6). Yet H3K9me3 and H3K27me3 signals remained diffuse as development proceeds in Spt16-1-depleted cells and no foci could be detected (Fig 6 and S8 Fig). Spt16-1 is thus not required for deposition of these silent histone marks but is required for foci formation, as reported for Pgm [24]. Altogether, our data suggest Spt16-1 acts downstream of (or parallel to) the scanRNA-directed heterochromatin pathway.

## Spt16-1 is essential for the correct nuclear localization of Pgm

In order to determine whether Spt16-1 acts up- or down-stream of Pgm during DNA elimination, we examined Pgm localization in *SPT16-1* KD cells by performing immunofluorescence experiments [39] during autogamy. Pgm progressively accumulates in the developing new MAC, with a peak corresponding to the stage when DNA DSBs are introduced at IES ends, and a decrease at later stages before it disappears from the developing MAC, as previously described [39]. Confocal microscopy indicated that Pgm accumulated in the developing new

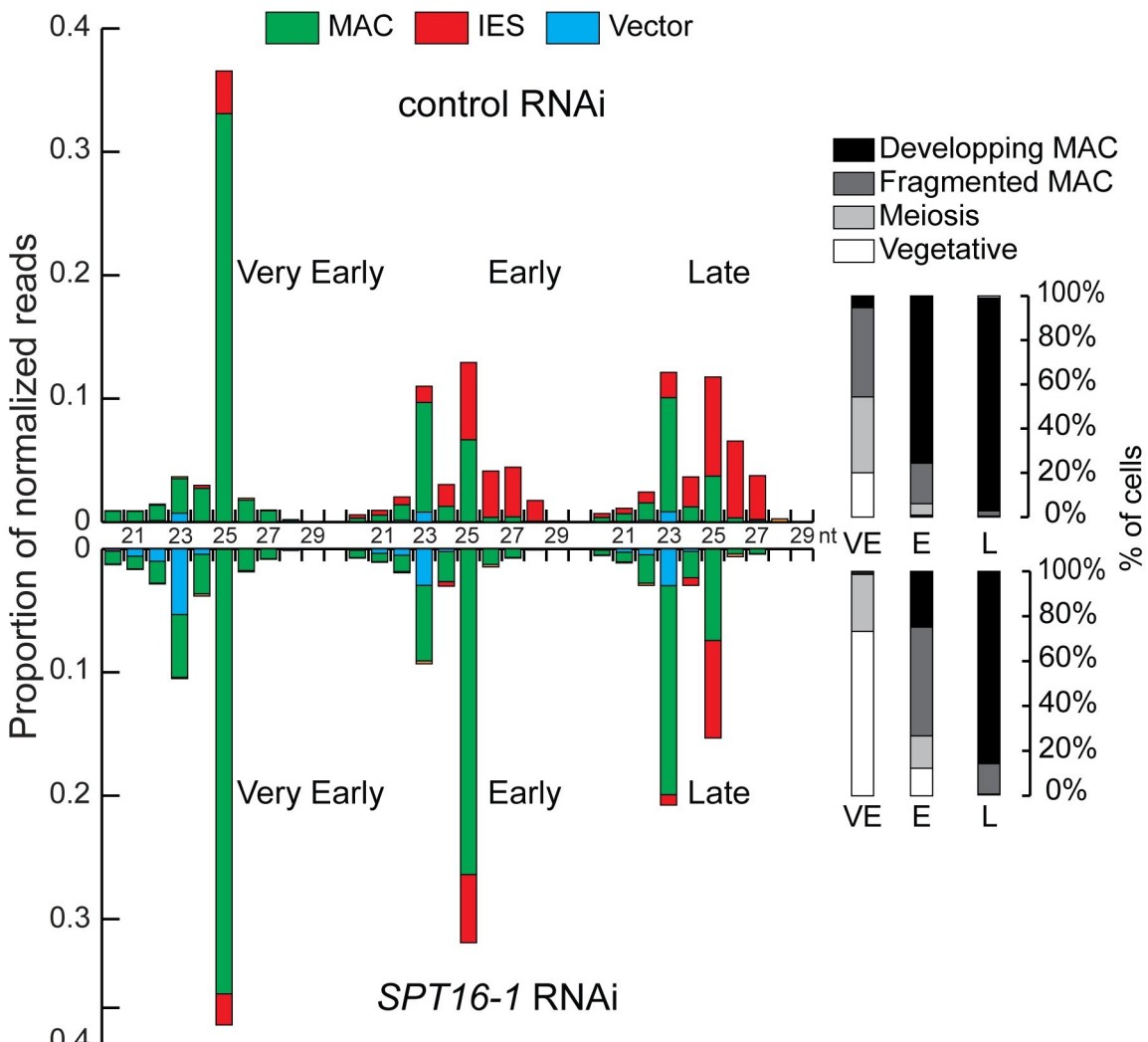

**Fig 5. Spt16-1 is required for iesRNA accumulation.** Analysis of small RNA populations in *SPT16-1* RNAi. Small RNA libraries at three different time-points during autogamy: very early (VE) (app. 20% of cell population undergo meiosis), early (E)(10 hours after the first time point), and late (L)(20 hours after the first time point), after control or *SPT16-1* RNAi were sequenced and mapped to the reference genomes (*Paramecium tetraurelia* MAC reference genome and MAC+IES reference genome). Bar plots show the normalized proportion of sRNA reads that match the MAC genome (green), annotated IESs (yellow) or the feeding vector (bue). The histograms display the cytological stages of the cell population at each developmental time point.

MAC at early stages of development (T5) in control cells (Fig 7A). In contrast, we could not detect Pgm accumulation in the developing new MAC in *SPT16-1* KD cells (Fig 7A). Estimation of nucleus volume indicated control and *SPT16-1* KD cell populations were at comparable developmental stages (S9 Fig). Quantification of Pgm fluorescence in the new developing MACs at this early stage of MAC development confirmed that the Pgm protein accumulated in the new MAC in control cells while it significantly diminished in the absence of Spt16-1 (Fig 7A) (see Materials and Methods), indicating that Spt16-1 is required for the accumulation of the Pgm protein in the developing new MAC. To rule out the possibility that Spt16-1 impacts the production of the Pgm protein itself, we performed Western blot analysis of total protein extracts at different time points during autogamy of control and Spt16-1-depleted cells, using Pgm antibodies. The Pgm protein was shown to progressively accumulate with a

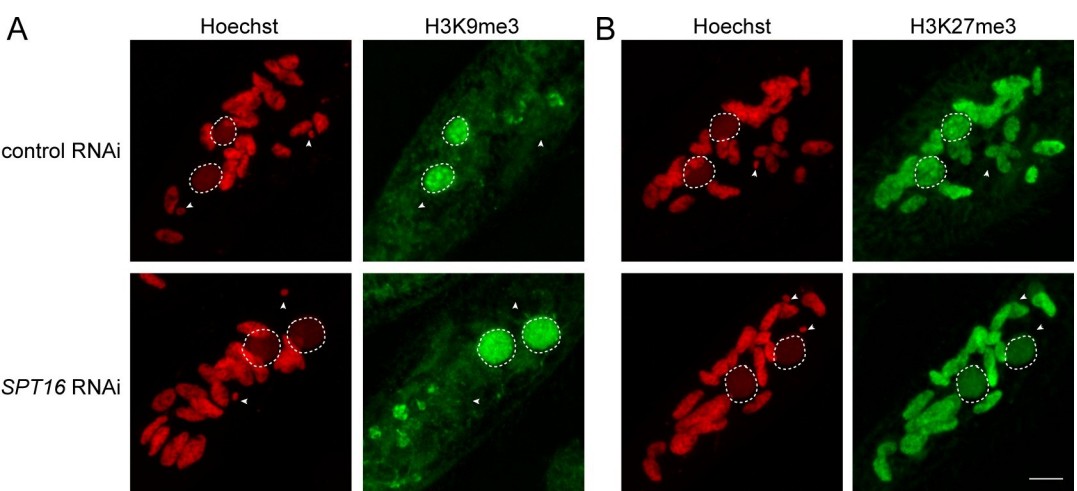

**Fig 6. Spt16-1 is not required for H3K9me3 and H3K27me3 deposition in the developing new macronuclei.** Z-projections of immunolabelling with (**A**) H3K9me3-specific antibodies (green) or (**B**) H3K27me3-specific antibodies and staining with Hoechst (red) in control or *SPT16-1* RNAi at early stages of development. Dashed white circles indicate the two developing MACs. White arrowheads indicate the MICs. The other Hoechst-stained nuclei are fragments from the maternal somatic MAC. Scale bar is 10 μm.

T5-T10 peak at similar levels in both control and *SPT16-1* KD cells (Fig 7B). Thus, Spt16-1 does not affect Pgm protein production, and instead, impairs its accumulation in the developing new MAC.

## Discussion

The histone chaperone Spt16, conserved in most eukaryotes [3], is generally encoded by a single gene. During evolution, two families of *SPT16* genes have emerged in *P. tetraurelia* (Fig 1). Family 1 consists of a single gene, *SPT16-1*. Family 2 comprises one to four genes, depending on the species. Although *Paramecium* have the most divergent group of Spt16 genes, they all contain the four canonical protein domains. In this study, we provide evidence for the functional specialization for Spt16-1. The three paralog genes Spt16-2a, -2b and -2c arising from *Paramecium* whole genome duplications [29], are constitutively expressed during vegetative growth and at all stages of the sexual cycle, and likely encode housekeeping Spt16 proteins. In contrast, Spt16-1 has a highly specialized expression pattern and a specific role during MAC development. The functional specialization of Spt16-1 is further supported by the conservation of an orthologous *SPT16-1* paralog in all species of the *P. aurelia* group, which undergo massive programmed genome rearrangements (S1 Fig).

We demonstrate that *SPT16-1* is required for programmed genome rearrangements. By genome resequencing upon *SPT16-1* KD, we show that Spt16-1 is necessary for all DNA elimination—the precise excision of IESs, the imprecise elimination of repeated sequences and transposable elements, and the maternal inheritance of gene deletions (Fig 4). There were no significant differences between the *SPT16-1* and *PGM* KD phenotypes and, in particular, no change in frequency or type of excision errors (S7 Fig). Previous work revealed the existence of partially overlapping pathways involved in IES excision [18,20,24,38,40]. Spt16 does not function in these specialized pathways and is instead a general factor that, like Pgm, is required for the elimination of all IESs [12].

The current model for elimination of the majority of MIC-specific sequences proposes that 25 nt-long scanRNAs produced by the entire MIC genome during meiosis are transferred to

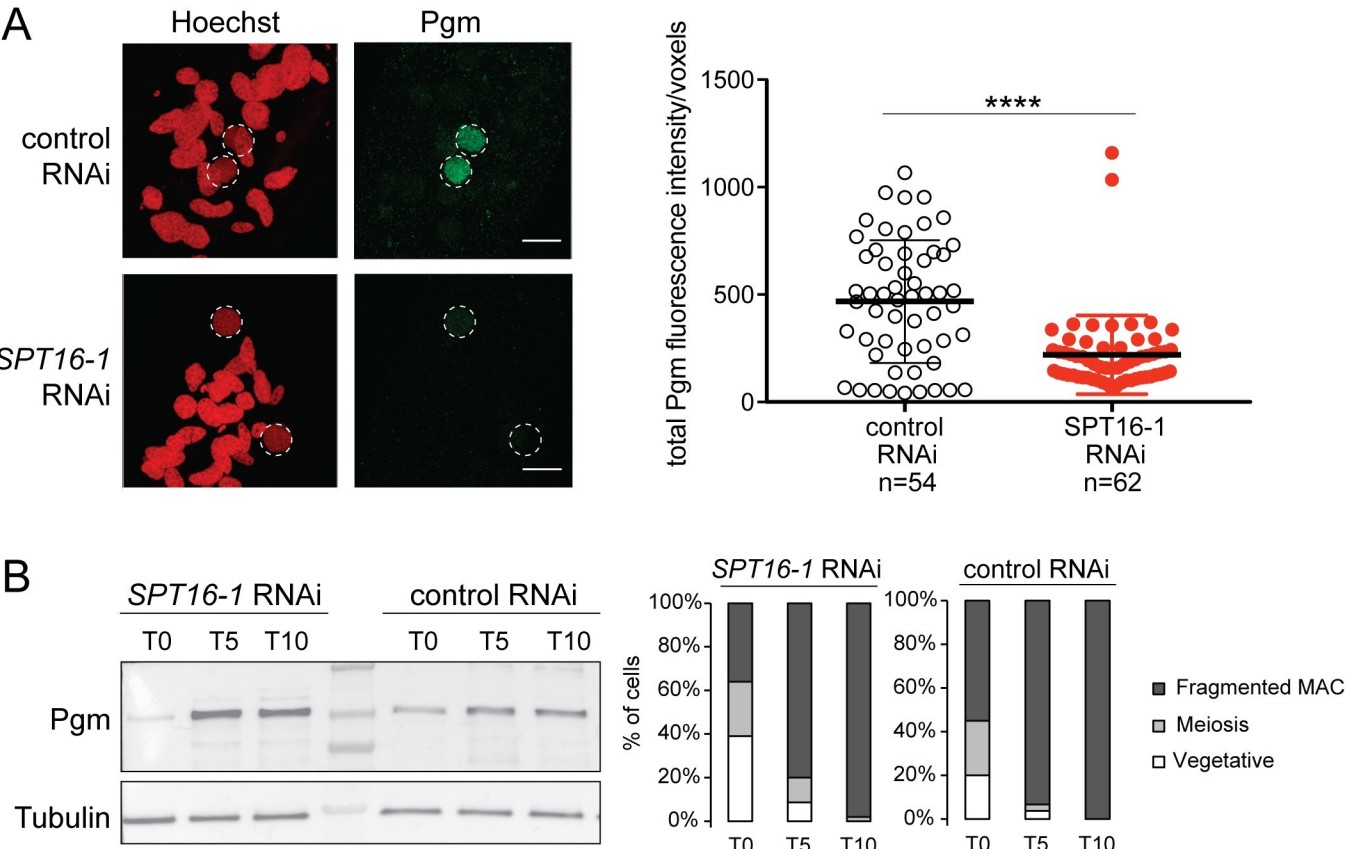

**Fig 7. Spt16-1 is required for Pgm nuclear localization. (A)** Z-projections of immunolabelling with Pgm antibodies (green) and staining with Hoechst (red) in control or *SPT16-1* RNAi at T5 during autogamy. Dashed white circles indicate the two developing MACs. The other Hoechst-stained nuclei are fragments from the maternal somatic MAC and the two germline MICs. Scale bar is 10 μm. Bar plots represent total Pgm fluorescence intensity in the developing new MAC divided by the number of voxels in both conditions (control or *SPT16-1* RNAi). **** for p<0.0001 in a Mann-Whitney statistical test. **(B)** Pgm expression during development in control or *SPT16-1* RNAi. Western blot of whole cell proteins extracted at T0, T5 and T10 during autogamy, when Pgm is normally detected by immunofluorescence, upon control or *SPT16-1* RNAi. Pgm antibodies were used for Pgm detection and alpha tubulin antibodies for normalization. The protein ladder (PageRuler Prestained Protein Ladder, 10 to 250 kDa, ThermoFisher Scientific) is shown. Cytology of the cell population at each developmental time point is displayed in the histograms.

the maternal MAC, where they are enriched in MIC-specific sequences, and then moved to the new MAC where they lead to the deposition of histone H3 post-translational modifications (H3K9me3 and H3K27me3) on homologous sequences, eventually triggering DNA elimination. High throughput sequencing of sRNA populations at different time points during sexual events and during MAC development showed that Spt16-1 is not required for scanRNA production nor for the selection process that allows enrichment in MIC-specific sequences (Fig 5). Consistent with Spt16-1 being essential for DNA elimination, 26–29 nt long iesRNAs, produced in the new MAC from excised IESs and apparently involved in IES excision [18,26], were not detected in Spt16-1-depleted cells. We further showed that Spt16-1 is not required for the deposition of H3K9me3 and H3K27me3 in the developing new MAC (Fig 6), similarly to Pgm [24,25]. Our analyses thus indicate that Spt16-1, like Pgm, acts independently of the scanRNA-mediated heterochromatin pathway.

Heterochromatin H3K27me3 and H3K9me3 marks normally display a dynamic pattern of localization during MAC development: the staining observed in immunofluorescence experiments with specific antibodies is first diffuse then reorganizes into nuclear foci that collapse into a single focus before it eventually disappears [24]. We previously showed that Pgm is

required for the formation of these nuclear bodies [24]. Similarly, we showed here that Spt16-1 was essential for their dynamic reorganization (Fig 6 and S8 Fig). The biological significance of the formation and aggregation of H3K9me3/H3K27me3 nuclear bodies remains unclear. This is reminiscent of phase separation, a phenomenon that gives rise to membraneless compartments that is suggested to be involved in heterochromatin formation [41,42]. The late timing of their formation relative to DNA elimination and their dependency toward Pgm and Spt16-1 factors suggest that H3K9me3/H3K27me3 nuclear bodies formation is an event that follows the introduction of DNA cleavages. We speculate that MIC-specific DNA associated with the factors needed for elimination (proteins and non-coding RNAs) are aggregated after excision from the chromosomes before degradation in the nucleus.

We showed that Spt16-1 is critical for localization of the Pgm protein in the developing new MAC (Fig 7) and, therefore, why DNA elimination is impaired in *SPT16-1* KD cells. Spt16-1 is not involved in the production or stability of the Pgm protein, as the levels of the Pgm protein are not affected by Spt16-1 depletion (Fig 7). Our quantitative image analysis of localization experiments showed a substantial reduction of the Pgm nuclear signal in the absence of Spt16-1 (Fig 7). Thus, Spt16-1 impacts Pgm localization, with Pgm presumably predominantly accumulating in the cytoplasm in the absence of Spt16-1. Our inability to detect cytoplasmic Pgm by immunolocalization may be explained by signal diffusion in the large cytoplasmic volume in *P. tetraurelia* cells (130 μm large, 30 μm width).

IES excision is initiated by the introduction of DNA double-strand breaks (DSBs) at IES ends, mediated by the Pgm endonuclease and assisted by Pgm-Like proteins [13,14]. The DSBs are repaired by the ligase IV- and Xrcc4-dependent classical non-homologous end joining (NHEJ) pathway [15]. We observed the absence of precise *de novo* IES excision junctions in *SPT16-1* KD cells (Fig 4), which could either reflect an inhibition of Pgm-dependent DNA cleavage or a problem in DNA repair. In Ligase IV-depleted *Paramecium* cells, the developing new MACs display a faint DAPI staining, likely due to lack of endoreplication caused by the accumulation of non-repaired DSBs [15]. This faint staining phenotype was not observed in the developing new MACs of *SPT16-1* or *PGM* KD cells (Fig 2), suggesting that Spt16-1 depletion inhibits DNA cleavage, as for Pgm. This conclusion is further supported by the impact of *SPT16-1* KD on Pgm localization. Pgm predominantly accumulates in the nucleus in normal conditions, while it does not in absence of Spt16-1 (Fig 7), explaining the lack of DNA elimination in *SPT16-1* KD cells and that Spt16-1 depletion phenocopies Pgm depletion.

Alternatively, Spt16-1 could play a role in the repair step following DSB generation by Pgm. FACT has been implicated in the recognition of DNA damage and in the reorganization of chromatin at damage sites to facilitate DNA repair [3]. FACT has been reported to interact with the Ku complex, an essential DSB repair factor that binds broken DNA ends and recruits downstream NHEJ proteins [43], and has been shown to remodel chromatin at repair sites [27]. In *P. tetraurelia*, a specialized Ku70/Ku80 heterodimer acts upstream of DNA cleavage, instead of downstream. Indeed, the depletion of the developmental-specific Ku80 paralog Ku80c abolished DNA cleavage at IES ends, resulting in retention of all IESs in the new MAC [16,44], as for Spt16-1 and Pgm depletion. Furthermore, similar to Spt16-1 being essential for Pgm nuclear localization, Ku80c is required for anchoring the Pgm endonuclease in the developing MAC [44]. Ku80c interaction with Pgm during MAC development is thought to license Pgm-dependent DNA cleavage. It is possible Spt16-1 functions by a similar mechanism, given the tight coupling between DNA cleavage and DSB repair during programmed DNA elimination in *Paramecium*. Further work is needed to examine whether Spt16-1 associates with the heterodimer Ku70/Ku80c and tethers the Pgm complex to chromatin, allowing Pgm to cleave DNA.

We propose that Spt16-1 mediates the interaction between Pgm and chromatin or DNA, either directly or indirectly, preventing Pgm export from the nucleus. Spt16 in other organisms is able to open chromatin and make the DNA accessible to the replication, transcription, or repair machineries [4,6,45]. Similarly, Spt16-1 could interact with histones and rearrange the chromatin to open it, so that Pgm and other excision machinery components can interact with DNA. FACT substantially increases the accessibility of nucleosomal DNA, by promoting formation, or stabilization, of a looser nucleosome structure, still bound to DNA but more likely to undergo reorganization through H2A/H2B displacement [7,8]. Chromatin remodeling mediated by Spt16-1 and occurring in Pgm-targeted regions may generate partially dissociated nucleosomes, leading to chromatin structures that are preferential substrates for Pgm-mediated DNA cleavages. Local Spt16-1-mediated remodeling of the chromatin could thus allow or promote both protein/protein and protein/DNA interactions within the targeted nucleosomes.

The FACT complex comprises two subunits, SPT16 and SSRP1/Pob3 [3]. We found three Pob3 homologs in the *P. tetraurelia* genome, one of which, *POB3-1*, displays the same expression profile as *SPT16-1*. RNAi-mediated *POB3-1* KD did not reveal any developmental phenotypes observed for Spt16-1 depletion (S3 Fig). As described in *S. pombe* [46], depletion of *Paramecium* Pob3-1 does not appear to be lethal. This observation might be explained by Spt16-1 and Pob3-1 not forming a FACT complex. Consistent with this hypothesis, the *Paramecium tetraurelia* Pob3 protein sequences (Pob3-1, as well as Pob3-2a and -2b) lack the N-terminal N domain, which is required for dimerization with Spt16 (S4 Fig) [32]. However, we cannot exclude that Spt16-1 and Pob3-1 interact indirectly. Like the *P. tetraurelia* Pob3 proteins, the Pob3 protein in the distantly related ciliate *Tetrahymena* lacks the N-terminal N dimerization domain (S4 Fig) and yet pull down experiments performed in *Tetrahymena* with the unique Spt16 protein retrieved the Pob3 protein [47]. Whether the *Tetrahymena* Spt16 and Pob3 proteins bind to each other directly or indirectly has not been tested. It is not known either whether these proteins play a role in programmed genome rearrangements that occur during *Tetrahymena* sexual cycle. Both the *P. tetraurelia* Spt16-1 and Pob3-1 proteins likely bind to histones. The aromatic and acidic amino acids required for interaction with histones are conserved in Spt16-1, suggesting it may act as a histone chaperone (S2 Fig). The histone binding domains are also present in Pob3, suggesting it may also function as a histone chaperone (S4 Fig). Further studies will be needed to determine whether Spt16-1 belongs to a heterodimer FACT complex, with Pob3-1, and acts as a histone chaperone, or acts alone, either as a histone chaperone, or in some other capacity, with some other activity.

In conclusion, our work establishes that a developmental-specific Spt16-1 is essential for the programmed genome rearrangements that occur during *Paramecium* development. There is evidence that histone chaperones can play an active role in the import of histones. Histone chaperones have also been shown to regulate the localization of chromatin modifying enzymes [48]. Here, we show that the Spt16-1 histone chaperone promotes the nuclear localization and activity of the Pgm endonuclease, revealing a mechanism to control nuclear availability of the enzymatic complex responsible for DNA cleavages that orchestrates genome reorganization.

## Materials and methods

### *Paramecium* strains, cultivation and autogamy

All experiments were carried out with the entirely homozygous wild type strain 51 of *P. tetraurelia* or with strain 51 cells deleted for genes A and ND7 in the macronucleus. Cells were grown in wheat grass powder (WGP) (Pines International) infusion medium bacterized the day before use with *Klebsiella pneumoniae* and supplemented with 0.8 mg/mL β-sitosterol (Merck). Cultivation and autogamy were carried out at 27˚C [49].

## Gene silencing experiments

Plasmids used for T7Pol-driven dsRNA production in silencing experiments were obtained by cloning PCR products from each gene using plasmid L4440 and *Escherichia coli* strain HT115 DE3, as previously described [33]. For *SPT16-1* silencing, two non-overlapping gene fragments covering positions 1704–2165 (SPT16_1#1) and 165–902 (SPT16_1#2) of PTET.51.1. G0710091 were used. The fragments used for *ND7* [50], *ICL7a* [21], *PGM-1* [13] and *EZL1-1* [24] are those previously published. Preparation of silencing medium and RNAi during autogamy was performed as described in [13]. Lethality of post-autogamous cells after silencing of *PGM* in GFP-Spt16-1 transformed cells was 90–100%. As expected, Pgm depletion led to retention of all IESs tested.

## Injection of GFP fusion transgenes

For the construction of in-frame *GFP-SPT16-1* fusion, a GFP coding fragment adapted to *Paramecium* codon usage was added by PCR fusion to the 5' end of the *SPT16-1* gene. As a result, the GFP is fused to the N-terminus of *SPT16-1* and the fusion protein is expressed under the control of the *SPT16-1* transcription signals (promoter and 3'UTR). It contains the 119-bp genomic region upstream of the *SPT16-1* open reading frame, the 308-bp genomic region downstream. The plasmid carrying the *GFP-SPT16-1* fusion transgene was linearized by *Rsr*II and microinjected into the MAC of vegetative cells. No lethality was observed in the post-autogamous progeny of injected cells, indicating that the GFP-SPT16-1 fusions did not interfere with normal progression of autogamy. For the construction of in-frame RNAi resistant *GFP-SPT16-1* fusion, the original *Nhe*I-*Swa*I restriction fragment of the *SPT16-1* coding sequence that comprises the RNAi target region *SPT16-1#2* was replaced by a synthetic DNA sequence (Eurofins Genomics). The fragment was designed to maximize nucleotide sequence divergence with the endogenous genomic locus without modifying the amino acid sequence of the encoded Spt16-1 protein. Plasmid carrying the RNAi-resistant *GFP-SPT16-1* fusion transgene was linearized by *Rsr*II and microinjected into the MAC of vegetative cells.

## DNA and RNA extraction and PCR, qPCR

DNA samples were typically extracted from 200-400-mL cultures of exponentially growing cells at <1,000 cells/mL or of autogamous cells at ≤2,000–4,000 cells/mL as previously described [51]. Small-scale DNA samples were prepared from 1,000 cells using the NucleoSpin Tissue kit (Macherey-Nagel). RNA samples were typically extracted from 200–400-mL cultures of exponentially growing cells at <1,000 cells/mL or of autogamous cells at 2,000–4,000 cells/ mL as previously described [51].

 PCR amplifications were performed in a final volume of 25 μL, with 10 pmol of each primer, 10 nmol of each dNTP and 1.9 U of Expand Long Template Enzyme mix (Expand Long Template PCR system, Roche). PCR products were analyzed on 0.8%-3% agarose gels. Oligonucleotides were purchased from Eurofins MWG Operon or SIGMA (see S2 Table).

## H3K9me3/H3K27me3 immunofluorescence

Cells were fixed as described in [24] with some modifications. Briefly, cells were collected in centrifuge tubes and fixed for 30 minutes in Solution I (10 mM EGTA, 25 mM HEPES, 2 mM MgCl2, 60 mM PIPES pH 6.9 (PHEM 1X); formaldehyde 1% (Sigma-Aldrich), Triton X-100 2.5%, Sucrose 4%), and for 10 minutes in solution II (PHEM 1X, formaldehyde 4%, Triton X-100 1.2%, Sucrose 4%). Following blocking in 3% bovine serum albumin (Sigma-Aldrich) supplemented Tris buffered saline (10 mM Tris pH 7.4, 0.15 M NaCl) -Tween 20 0.1% (TBST) for

10 minutes, fixed cells were incubated overnight at room temperature with primary antibodies: rabbit anti-H3K9me3 (07–442, Millipore; 1:200), rabbit anti-H3K27me3 (07–449, Millipore; 1:500). After two washes in TBST 3% BSA, cells were labeled with Alexa Fluor 568-conjugated goat anti-rabbit IgG (Invitrogen, catalog number #A-11036, 1:500) for 1 h, stained with 1 μg/mL Hoechst for 5–10 minutes, washed in TBST and finally mounted in Citifluor AF2 glycerol solution (Citifluor Ltd, London).

## Pgm immunofluorescence

Pgm Immunofluorescence was performed as described in [39]. Briefly, cells were permeabilized for 4 minutes in PHEM 1X, Triton X-100 1% and fixed for 15 minutes in PHEM 1X, paraformaldehyde 2%, then washed twice in TBST (10 mM Tris pH 7.4, 0.15 M NaCl, 0.1% Tween 20) 3% BSA (bovine serum albumin, Sigma-Aldrich). Cells were incubated for 2 h at room temperature with anti-Pgm 2659 primary antibodies (1:300 in Signal + solution A for immunostaining, GenTex), and washed with TBST 3% BSA prior to 40 min incubation with Alexa fluor 488 goat anti-rabbit IgG secondary antibody (1:300, ThermoFisher Scientific), followed by DAPI staining (0.2 μg/mL) for 5 minutes and finally mounted in Citifluor AF2 glycerol solution (Citifluor Ltd, London).

## Fixation of GFP-Spt16 injected cells

Cells were fixed and stained for 10 minutes in PHEM 1X (10 mM EGTA, 25 mM HEPES, 2 mM MgCl2, 60m M PIPES pH 6.9), paraformaldehyde 2%, Hoechst 1 μg/mL, washed in 1X PBS and finally mounted in Citifluor AF2 glycerol solution (Citifluor Ltd, London).

## Image acquisition and quantification

All images were acquired using a Zeiss LSM 710 laser-scanning confocal microscope and a Plan-Apochromat 63x/1.40 oil DIC M27 objective. Z-series were performed with Z-steps of 0.39 μm (or 0.5 μm for quantification). Quantification was performed using ImageJ. For each cell, the volume of the nucleus (in voxels) was estimated as follows: using the Hoechst channel, the top and bottom Z stacks of the developing MAC were defined to estimate nucleus height in pixels. The equatorial Z stack of the developing MAC was defined and the corresponding developing MAC surface was measured in pixels. The estimated volume of the developing MAC was then calculated as the product of the obtained nucleus height by the median surface. For each Z stack of the developing MAC, the Pgm fluorescence intensity was measured and corrected using the ImageJ "subtract background" tool. The sum of the corrected Pgm fluorescence intensities for all the Z stacks, which corresponds to the total Pgm fluorescence intensity, was divided by the estimated volume to obtain the Pgm fluorescence intensity per voxel in each nucleus.

## Whole cell protein extracts and Western-Blot analysis

Cell pellets from autogamous cells at ≤2,000–4,000 cells/mL were frozen at -80˚C in liquid nitrogen. Pellets were lysed 3 min in boiling SDS 5% and Protease Inhibitor Cocktail Set I—Calbiochem 5X (Merck) and centrifuged at maximum speed. Supernatant with proteins were collected and Laemmli buffer was added. 10 μg of protein extracts were used for Western blot. Electrophoresis and blotting were carried out according to standard procedures. The Pgm (1:500; Proteogenix 2659-Guinea Pig) primary antibody was used [39]. Secondary horseradish peroxidase-conjugated donkey anti-rabbit IgG antibody (Promega) was used at 1:5000 dilution followed by detection by ECL (SuperSignal West Pico Chemiluminescent Substrate, Thermo Scientific). For

normalization, the membranes probed with Pgm antibody were stripped in stripping buffer (ThermoScientific) and probed against with alpha-Tubulin antibody (TEU435 [39]).

## DNA and sRNA sequencing

DNA for deep-sequencing was isolated from post-autogamous cells as previously described [12] and sequenced by a paired-end strategy using Illumina GA-IIx and Hi-Seq next-generation sequencers. Accession numbers and sequencing data are displayed in S3 Table.

Purification, sequencing and analysis of sRNAs from control and Spt16-depleted cells were carried out as previously described [40]. Briefly, for sRNA library construction, the ∼19–28 nt fraction was purified from total RNA by polyacrylamide gel electrophoresis (15%, 19:1 acrylamide:bisacrylamide) and gel-eluted with 0.3 M sodium chloride, followed by ethanol precipitation. The eluate was used for library construction using standard Illumina protocols. The 20–30 nt sRNA reads were filtered for known contaminants (*Paramecium* rDNA, mitochondrial DNA, feeding bacteria genomes and L4440 feeding vector sequences) (S4 Table). In addition, the 23 nt siRNA reads that map to the RNAi targets were removed. The filtered reads were mapped to reference MAC and MAC+IES genomes. Read counts were normalized using the total number of filtered reads.

## Reference genomes

The following reference genomes [12] were used in the IES analyses and for read mapping.

Paramecium tetraurelia strain 51 MAC genome (v1.0): https://paramecium.i2bc.paris-saclay.fr/download/Paramecium/tetraurelia/51/sequences/ptetraurelia_mac_51.fa

Paramecium tetraurelia strain 51 MAC+IES genome (v1.0): https://paramecium.i2bc.paris-saclay.fr/download/Paramecium/tetraurelia/51/sequences/ptetraurelia_mac_51_with_ies.fa

Paramecium tetraurelia strain 51 PGM contigs (v1): https://paramecium.i2bc.paris-saclay.fr/download/Paramecium/tetraurelia/51/sequences/contigs_ABK_COSP_best_k51_no_scaf.fa

Paramecium tetraurelia strain 51 MIC contigs (v0): https://paramecium.i2bc.paris-saclay.fr/download/Paramecium/tetraurelia/51/sequences/ptetraurelia_mic2.fa

Macronuclear DNA reads for PiggyMac depleted cells were obtained from the European Nucleotide Archive (Accession number ERA137420) (PGM).

## Genome-wide analysis of IES retention

Analysis was made using the multi-threaded Perl software PARTIES available at https://github.com/oarnaiz/ParTIES [36].

## Supporting information

**S1 Fig. Identification of Spt16 homologues in *P. tetraurelia*.** Phylogenetic tree of Spt16 proteins from *P. tetraurelia*, other sequenced *Paramecium species* and other eukaryotes based on alignment of full length protein sequence with MUSCLE. Phylogeny of the alignment was generated using PhyML 3.0 on Phylogeny.fr with bootstrapping procedure using 100 bootstraps. Tree has been viewed using Tree.Dyn 198.3. Accession numbers are indicated in S1 Table. (TIF)

**S2 Fig. Spt16 is a conserved protein among eukaryotes.** MUSCLE alignment of Spt16 homologues from *P. tetraurelia* and other eukaryotes. Domains idientified by Interpro are indicated in colors. Conserved aromatic and acidic residues in MBD are shown with asterisk (*). Conserved residues are highlighted in black and grey (amino acids with same physical and chemical properties). P.t. *Paramecium tetraurelia*, T.t *Tetrahymena thermophila*, S.p.

*Schyzosaccharomyces pombe*, S.c. *Saccharomyces cerevisiae*, C.e. *Caenorhabditis elegans*, D.m. *Drosophila melanogaster*, H.s. *Homo sapiens*, M.m. *Mus musculus*, A.t. *Arabidopsis thaliana*.
(PDF)

**S3 Fig. *Paramecium tetraurelia* Pob3 proteins.** **(A)** Phylogenetic tree of Pob3 proteins from *Paramecium tetraurelia* (Pt), *Tetrahymena thermophila* (Tt), *Saccharomyces cerevisiae* (Sc), *Schizosaccharomyces pombe* (Sp), *Caenorhabditis elegans* (Ce), *Drosophila melanogaster* (Dm), human (Hs) and *Arabidopsis thaliana* (At) based on the alignment of full-length protein sequences. The tree was generated with PhyML 3.0 with bootstrapping procedure and visualized with Tree.Dyn 198.3. Accession numbers are provided in S1 Table. **(B)** Conserved domains (colored boxes) in *P. tetraurelia* Pob3-1 protein. **(C)** *POB3* gene expression profiles during the life cycle. Mean mRNA expression levels were determined by RNA sequencing during vegetative growth and at different time points during autogamy [31]. **(D)** Production of post-autogamous sexual progeny following *POB3-1* gene silencing. The gene targeted in each silencing experiment is indicated. Two non-overlapping silencing fragments (#1: positions 36–420 and #2: positions 442–878 of PTET.51.1.G0610231) of the *POB3-1* gene were used independently. The *ND7* or *ICL7* genes were used as control RNAi targets, since their silencing has no effect on sexual processes [21]. The total number of autogamous cells analyzed for each RNAi and the number of independent experiments (in parenthesis) are indicated. The absence of lethality observed after *POB3-1* KDs should be taken with caution as the level of KDs was not measured. **(E)** PCR analysis of IES retention. Primers (black arrows, S2 Table) are located on either side of the IES. Total DNA samples were prepared from autogamous cell population upon RNAi-mediated silencing of the indicated genes. Because the maternal MAC is still present at this stage, the excised version is amplified in all cases; the IES-retaining fragment can be detected only if it accumulates in the zygotic developing MACs.
(PDF)

**S4 Fig. Multiple alignment of Pob3 proteins.** CLUSTAL Omega alignment of Pob3 homologues from *P. tetraurelia* and other eukaryotes. Domains identified by Interpro are indicated in colors. The Pob3M and Pob3C domains contain two PH motifs, similar to SPT16 M domain, which are likely to be involved in binding H3/H4 due to the strong similarity with the dual PH motifs of Rtt106, a known H3/H4 chaperone. *Paramecium* and *Tetrahymena* Pob3 proteins lack the Pob3_N domain. The Pob3_N domain is required for dimerization with SPT16. Residues important for interaction of Sc_SSRP1-CTD with H2A-H2B dimers are shown with pink squares [52]. P.t. *Paramecium tetraurelia*, T.t *Tetrahymena thermophila*, S.p. *Schyzosaccharomyces pombe*, S.c. *Saccharomyces cerevisiae*.
(TIF)

**S5 Fig. *SPT16-1* mRNA expression levels during autogamy.** The expression levels of *SPT16-1* mRNA are measured using normalized read counts (RPM, Reads per million mapped reads) at three time points (Vegetative (Veg), T0 and T10) during autogamy upon *SPT16-1* (solid line) and control (dashed line) RNAi. Only the nucleotides outside of the targeted RNAi region within the *SPT16-1* gene are considered.
(PDF)

**S6 Fig. The GFP-Spt16-1 is functional.** Overlay of Z-projections of magnified views of GFP-Spt16-1 (green) and Hoechst (red) in *SPT16-1* RNAi during MAC development. Dashed white circles indicate the two developing MACs. The other Hoechst-stained nuclei are fragments from the maternal somatic MAC. Scale bar is 10 μm.
(TIF)

**S7 Fig. *SPT16-1* KD results in retention of all IESs and does not lead to increased excision errors. (A)** Excision of all IESs is altered upon *SPT16-1* KD. 98.6% (44,334) IESs are statistically significantly retained in the developing MAC after *SPT16-1* KD, similar to the 99% (44,491) observed after *PGM* KD (sample from [12])). **(B)** Quantification of excision errors between control (no RNAi), *SPT16-1* and *PGM* RNAi (FACS-sample from [11]). **(C)** Quantification and identification of error types are made with the PARTIES software described in [36]. There is no significant difference between the three conditions.
(TIF)

**S8 Fig. Spt16-1 is required for H3K9me3 and H3K27me3 foci formation in new developing MACs.** Z-projections of immunolabelling with **(A)** H3K9me3-specific antibodies (green) or **(B)** H3K27me3-specific antibodies and staining with Hoechst (red) in control or *SPT16-1* RNAi at late stages of development. Dashed white circles indicate the two developing MACs. White arrowheads indicate the MICs. The other Hoechst-stained nuclei are fragments from the maternal somatic MAC. Scale bar is 10 μm.
(TIF)

**S9 Fig. Estimated nucleus volume in the developing new MAC.** The Pgm fluorescence intensity is plotted as a function of the estimated nucleus volume in voxels in control or *SPT16-1* KD cells (same data as Fig 7A). The estimated volume of the developing macronucleus increases as development progresses in control and *SPT16-1* KD cells.
(TIF)

**S1 File.** Related to legends of Fig 1, Fig 2 and Fig 3.
(DOCX)

**S2 File. Numerical data.**
(XLSX)

**S1 Table. Accession number of proteins used in the phylogenetic analyses.** Accession number, name of the protein and species are given. *Paramecium* database: https://paramecium.i2bc.paris-saclay.fr/, *T. thermophila* database: http://ciliate.org/index.php/home/welcome, *S. pombe* database: https://www.pombase.org/, *S. cerevisiae* database: https://www.yeastgenome.org/, *C. elegans* database: https://wormbase.org/#012-34-5, *D. melanogaster* database: http://flybase.org/, *M. musculus* database: http://www.informatics.jax.org/, *H. sapiens* database: https://www.genenames.org, *A. thaliana* database: https://www.arabidopsis.org/.
(DOCX)

**S2 Table. Oligonucleotides used in this study.**
(DOCX)

**S3 Table. Sequencing and mapping statistics for DNAseq.** Read statistics are provided for the *SPT16-1* RNAi sample sequenced for this study.
(DOCX)

**S4 Table. Description of sRNA-seq data.** Read statistics are provided for the control RNAi and *SPT16-1* RNAi samples sequenced for this study.
(DOCX)

## Acknowledgments

We wish to thank the members of the Duharcourt lab for fruitful discussions and Caridad Miró Pina for her help in Western blot analysis. We are grateful to Vinciane Régnier and

Mireille Bétermier for sharing the Pgm antibodies. We thank Vincent Contremoulins for his help with image analysis. We thank Sophie Polo for critical reading of the manuscript. The sequencing benefited from the facilities and expertise of the high-throughput sequencing platform of I2BC.

## Author Contributions

**Conceptualization:** Augustin de Vanssay, Amandine Touzeau, Sandra Duharcourt.

**Data curation:** Olivier Arnaiz, Sandra Duharcourt.

**Formal analysis:** Augustin de Vanssay, Amandine Touzeau, Olivier Arnaiz, Andrea Frapporti.

**Funding acquisition:** Sandra Duharcourt.

**Investigation:** Augustin de Vanssay, Amandine Touzeau, Olivier Arnaiz, Andrea Frapporti, Jamie Phipps.

**Methodology:** Augustin de Vanssay, Amandine Touzeau, Olivier Arnaiz, Andrea Frapporti, Jamie Phipps, Sandra Duharcourt.

**Project administration:** Sandra Duharcourt.

**Resources:** Olivier Arnaiz.

**Software:** Olivier Arnaiz.

**Supervision:** Sandra Duharcourt.

**Validation:** Augustin de Vanssay, Amandine Touzeau, Olivier Arnaiz, Sandra Duharcourt.

**Visualization:** Augustin de Vanssay, Amandine Touzeau, Olivier Arnaiz, Andrea Frapporti, Sandra Duharcourt.

**Writing – original draft:** Sandra Duharcourt.

**Writing – review & editing:** Olivier Arnaiz, Andrea Frapporti, Sandra Duharcourt.

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
