## [Decision Letter · Decision Letter 0]

24 Apr 2020

Dear Dr Duharcourt,

Thank you very much for submitting your Research Article entitled 'The Paramecium histone chaperone Spt16-1 is required for Pgm endonuclease function in programmed genome rearrangements' to PLOS Genetics. Your manuscript was fully evaluated at the editorial level and by independent peer reviewers. The reviewers appreciated the attention to an important problem, but raised some substantial concerns about the current manuscript. Based on the reviews, we will not be able to accept this version of the manuscript, but we would be willing to review again a much-revised version. We cannot, of course, promise publication at that time. We note that we are aware that significant experimental work may not be currently be possible in light of Covid-19 but a significant fraction of the very constructive suggestions by the reviewers pertain to the writing and an expanded discussion of the results and their implications; we feel confident that you may be able to address most of these with a careful editorial revision.

If you decide to revise the manuscript for further consideration at PLOS Genetics, please aim to resubmit within the next 60 days, unless it will take extra time to address the concerns of the reviewers, in which case we would appreciate an expected resubmission date by email to plosgenetics@plos.org.

[LINK]

We are sorry that we cannot be more positive about your manuscript at this stage. Please do not hesitate to contact us if you have any concerns or questions.

Yours sincerely,

Harmit S. Malik

Associate Editor

PLOS Genetics

Wendy Bickmore

Section Editor: Epigenetics

PLOS Genetics

Reviewer's Responses to Questions

**Comments to the Authors:**

Reviewer #1: In this manuscript, the authors clearly show that SPT16-1 is required for programmed genome rearrangement in Paramecium and this requirement is explained by the requirement SPT16-1 in the macronuclear (MAC) localization of Pgm, the enzyme responsible for DNA excision in programmed genome rearrangement. The data presented in this manuscript are explained clearly and convincing. However, the two following points make me reluctant to fully recommend the manuscript for publication in PLoS Genetics in the current form.

1) It is not clear whether SPT16-1 act as a part of FACT histone chaperone in programmed genome rearrangement.

Because Spt16-1 is evolutionary diverged (Figure 1), it might no longer be a histone chaperone but acquired some special function. Indeed, in Discussion, the authors mentioned that one of the Pob3 homologs in Paramecium, which shows a similar expression pattern with Spt16-1, is not required for programmed genome rearrangement. This result may indicate that Spt16-1 acts in programmed genome rearrangement not as a histone chaperone.

Does Spt16-1 forms FACT with with Pob3/SSRP1? Does Spt16-1 show any sign of histone chaperone activity? Does the loss of Spt16-1 alter chromatin architecture (such as nucleosome density, positioning, or modifications)? Especially, it is interesting (and doable) to investigate accumulations and localizations of H3K9me and H3K27me, given the fact that FACT promotes accumulation of Swi6 at heterochromatin independently of RNAi (Lejeune et al 2007).

2) It is not clear whether the requirement of SPT16-1 in programmed genome rearrangement is direct or indirect.

Although it is possible that “SPT16-1 is specifically required for programmed genome rearrangements” as the authors concluded (page 9), it is also possible that Spt16-1 has some more general function in the differentiation of the MAC, which eventually indirectly promotes genome rearrangements. For example, Spt16-1 may be required for activating the zygotic gene expressions in the MAC, and zygotic expression of some gene may be essential for the proper nuclear localization of Pgm.

Are PGML genes, which are known to be required for the MAC localization of Pgm (Bischerour et al., 2018), expressed normally in the absence of Spt16-1? Are there any other measurable processes of the MAC differentiation, which the authors can check if they are unaffected in SPT16-1 RNAi?

Other minor points:

3) Page 6, line 7: Why viability of non-injected SPT16-1 RNAi cells is much higher than those in the experiments shown in Figure 2A?

4) Because the overall retention scores are much lower in PGM RNAi than those in SPT16-1 RNAi in Figure 4D, it seems eliminations of IESs are more mildly affected in SPT16-1 RNAi than in PGM RNAi. Is this because Pgm is still active in SPT16-1 RNAi and can catalyze some IES eliminations? If so, why no IES elimination was detected in SPT16-1 RNAi in Figure 4B and C? I believe some explanations are necessary for this point.

5) I think, for general readers, it is difficult to understand how the authors concluded that the selection of scanRNA is normal and iesRNA is reduced in Spt16-1 RNAi from the data shown in Figure 5. I think much more explanations are necessary for how to interpret the data.

Reviewer #2: In this manuscript entitled “The Paramecium histone chaperone Spt16-1 is required for Pgm endonuclease function in programmed genome rearrangements”, the authors identify four Spt16-like histone chaperone homologues encoded in the genome of the ciliate Paramecium tetraurelia. These four can be classified into two gene families. They investigate the function of the most divergent one, Spt16-1, which is expressed exclusively during sexual development when it localizes specifically to developing somatic macronuclei. RNAi knockdown of this protein results in loss in the ability of cells to produce sexual progeny and failure to execute programmed elimination of germline-limited DNA. The inhibition of this major reorganization of the developing somatic genome is associated with the failure of the PiggyMac endonuclease, which initiates DNA elimination, to accumulate in the developing nucleus. Small RNAs that are required for DNA elimination accumulate normally, but a second class of small RNAs (iesRNAs) that are associated with eliminated sequences post excision do not accumulate. These finding suggests that Spt16-1 is required to ensure that this endonuclease is able to associate with the developing genome to initiate DNA elimination. The finding that an Spt16 homolog is required for DNA elimination further implicates chromatin-based regulation in this massive genome restructuring.

Strengths of the Study:

This well-executed study reports the discovery and characterization of a novel histone chaperone and shows that it has evolved to specifically regulate programmed DNA elimination in Paramecium. The data presented are quite clear and well-controlled. The RNAi inhibition of Spt16-1 expression appears to be very penetrant, leading to strong phenotypes allowing for robust conclusions. The authors show that Spt16-1 knockdown blocks all DNA elimination, and leads to a failure of Pgm accumulation in the developing nucleus, results that show that it acts after scanRNA production but concurrently or before Pgm action. This study, along with the senior author’s previous work showing a role for histone methylation in transposon silencing and DNA elimination, further highlights a role for chromatin structure in guiding genome reorganization. It has the added novelty of showing that Spt16 related proteins can undergo specialization after gene duplication and adopt specific roles in genome regulation.

Weaknesses of the study:

Major comments:

1. Although the study was well-executed, the presentation of the manuscript, including the introduction and results were rather narrowly focused on the role of Spt16-1 in programmed DNA rearrangements in Paramecium. The discovery that a specialized Spt16 exists potentially has broad implications, but the authors left speculation almost fully to the readers’ imaginations. Given the general audience of Plos Genetics, I would like to see the implications of the study made more explicit to the readers. The authors did discuss the relationship between Spt16 more generally in DNA repair, but the parallels in function are not explored. In the canonical case, Spt16 would work after damage, but their data suggest that it acts before breaks would be induced. Thus this reviewer found the discussion a bit superficial.

2. In trying to connect the role of Spt16 in DNA damage and DNA elimination, is it possible that Spt16-1 interacts with the Paramecium KU complex? Could they act together in the recruitment of Pgm to sites of DNA breaks? If they have similar phenotypes, it would be interesting to test whether Spt16-1 is part of the KU complex or helps target KU to chromatin.

3. What is the status of histone H3K9 and K27 methylation upon Spt16-1 knockdown? Given that this research group previously showed that “… development specific H3K27me3 and H3K9me3 ensure specific demarcation of very short germline sequences from the adjacent somatic sequences…” (from abstract of ref. #24), it sounds like they are proposing a similar role for Spt16. By not examining the status of H3 methylation, the authors have not informed the readers of the possible processes that might be affected by loss of Spt16-1 function.

Minor comments and text modifications:

1. Line 3 of introduction, suggestion: add “subsequent” to the sentence “…heterochromatin formation and subsequent DNA excision and repair...”

2. Introduction, 2nd Paragraph: Flip first and second sentences. Discussing the role of the macronucleus after introducing micronuclei, then starting again to describe germline functions disrupts the information flow.

3. Bottom of page 3: Typo ”sequences” also, rewrite the sentence. Histone H3 ptm are deposited on chromatin associated with sequences to be eliminated during development. If sequences have been eliminated (past tense) then they are not there to have their associated chromatin modified.

4. Page 5 Line 7: missing “growth” “…during vegetative growth (Figure 1C)….”

5. Page 6 Line 5: the statement: “… as no lethality was observed in the post-autogamous progeny of injected cells…” is a bit stronger than the data show. The author’s saw nearly full rescue of the RNAi-induced lethality, but it is still a bit lower than wild-type. It is accurately described lower down in the paragraph. The authors should modify the text to reflect the data.

6. In a few places, the authors used the word “if” when then should have used “whether”. “If” should be used in sentences that have the grammatical structure: if this, then that. When the sentence starts: “…to determine..” use “whether”, which means “to determine whether or not” something is correct.

7. Page 8 paragraph 2 line 5: typo, “it” not “its”

**Have all data underlying the figures and results presented in the manuscript been provided?**

Reviewer #1: Yes

Reviewer #2: Yes

PLOS authors have the option to publish the peer review history of their article (what does this mean?). If published, this will include your full peer review and any attached files.

Reviewer #1: No

Reviewer #2: No

---

## [Decision Letter · Decision Letter 1]

24 Jun 2020

Dear Dr Duharcourt,

We are pleased to inform you that your manuscript entitled "The Paramecium histone chaperone Spt16-1 is required for Pgm endonuclease function in programmed genome rearrangements" has been editorially accepted for publication in PLOS Genetics. Congratulations!

There are two outstanding comments from Reviewer #1 to consider- one concerning the title and whether this is a true Spt6 and the second regarding the presentation format for Figure 4D. The editor concurs with the reviewer on the second, but not the first suggestion. The Venn diagram suggested for Figure 4D may be preferred or would be a nice supplementary figure in addition. However, we leave both of these changes to your discretion in the final uploaded version.

Yours sincerely,

Harmit S. Malik

Associate Editor

PLOS Genetics

Wendy Bickmore

Section Editor: Epigenetics

PLOS Genetics

Comments from the reviewers (if applicable):

Reviewer's Responses to Questions

**Comments to the Authors:**

Reviewer #1: I understand that the authors’ arguments that investigations of the biochemical activity of Spt16-1 and nucleosome/histone localization are technically difficult. Then, I have a suggestion: Because there is no evidence that Spt16-1 acts as a histone chaperone, I believe it should be called as “putative” or “potential” histone chaperone in the manuscript. For example, title might better be “A Paramecium homolog of histone chaperone Spt16 is ..” or “The Paramecium putative histone chaperone Spt16-1 is ..“. Main text is also better to be changed accordingly.

I still have a comment for Fig 4D: If “Absolute values of RS cannot be compared from one knockdown to the other” why do the authors show a figure comparing RSs of the different knockdown experiments? If they just want to compare which IESs are significantly affected, showing it by a Venn diagram would be better.

Reviewer #2: Overview (copied from review of original manuscript)

In this manuscript entitled “The Paramecium histone chaperone Spt16-1 is required for Pgm endonuclease function in programmed genome rearrangements”, the authors identify four Spt16-like histone chaperone homologues encoded in the genome of the ciliate Paramecium tetraurelia. These four can be classified into two gene families. They investigate the function of the most divergent one, Spt16-1, which is expressed exclusively during sexual development when it localizes specifically to developing somatic macronuclei. RNAi knockdown of this protein results in loss in the ability of cells to produce sexual progeny and failure to execute programmed elimination of germline-limited DNA. The inhibition of this major reorganization of the developing somatic genome is associated with the failure of the PiggyMac endonuclease, which initiates DNA elimination, to accumulate in the developing nucleus. Small RNAs that are required for DNA elimination accumulate normally, but a second class of small RNAs (iesRNAs) that are associated with eliminated sequences post excision do not accumulate. These finding suggests that Spt16-1 is required to ensure that this endonuclease is able to associate with the developing genome to initiate DNA elimination. The finding that an Spt16 homolog is required for DNA elimination further implicates chromatin-based regulation in this massive genome restructuring.

Strengths of the Study (copied from review of original manuscript):

This well-executed study reports the discovery and characterization of a novel histone chaperone and shows that it has evolved to specifically regulate programmed DNA elimination in Paramecium. The data presented are quite clear and well-controlled. The RNAi inhibition of Spt16-1 expression appears to be very penetrant, leading to strong phenotypes allowing for robust conclusions. The authors show that Spt16-1 knockdown blocks all DNA elimination, and leads to a failure of Pgm accumulation in the developing nucleus, results that show that it acts after scanRNA production but concurrently or before Pgm action. This study, along with the senior author’s previous work showing a role for histone methylation in transposon silencing and DNA elimination, further highlights a role for chromatin structure in guiding genome reorganization. It has the added novelty of showing that Spt16 related proteins can undergo specialization after gene duplication and adopt specific roles in genome regulation.

Comments on revised manuscript:

After reading the revised manuscript and the authors' responses to the previous two reviewers, the authors have address my major concerns about weaknesses in this manuscript. The addition of histone methylation data provides a key piece of information previously missing, and the observation that methylation occurs, but that it remains dispersed in the developing nucleus provides further support for their conclusion that Spt16-1 helps direct cleavage of IESs from the genome. The authors shared gene expression data with the reviewers that support the idea that Spt16-1 knockdown effects are not likely due to indirect effects on expression of genes essential of DNA elimination. The additional data and the modification to the discussion have made this study one that should interest a variety of Plos Genetics readers.

**Have all data underlying the figures and results presented in the manuscript been provided?**

Reviewer #1: Yes

Reviewer #2: Yes

PLOS authors have the option to publish the peer review history of their article (what does this mean?). If published, this will include your full peer review and any attached files.

Reviewer #1: No

Reviewer #2: **Yes: **Douglas Chalker

**Data Deposition**

http://datadryad.org/submit?journalID=pgenetics&manu=PGENETICS-D-20-00395R1

**Press Queries**

---

## [Editor Report · Acceptance letter]

15 Jul 2020

PGENETICS-D-20-00395R1 

The Paramecium histone chaperone Spt16-1 is required for Pgm endonuclease function in programmed genome rearrangements 

Dear Dr Duharcourt, 

We are pleased to inform you that your manuscript entitled "The Paramecium histone chaperone Spt16-1 is required for Pgm endonuclease function in programmed genome rearrangements" has been formally accepted for publication in PLOS Genetics! Your manuscript is now with our production department and you will be notified of the publication date in due course.

With kind regards,

Jason Norris

PLOS Genetics

On behalf of:
